# Comparison of Endoscopic Hemostasis for Endoscopic Sphincterotomy Bleeding between a Novel Self-Assembling Peptide and Conventional Technique

**DOI:** 10.3390/jcm12010079

**Published:** 2022-12-22

**Authors:** Yuki Uba, Takeshi Ogura, Saori Ueno, Atsushi Okuda, Nobu Nishioka, Akira Miyano, Yoshitaro Yamamoto, Kimi Bessho, Mitsuki Tomita, Junichi Nakamura, Akitoshi Hakoda, Hiroki Nishikawa

**Affiliations:** 12nd Department of Internal Medicine, Osaka Medical and Pharmaceutical University, Osaka 565-0871, Japan; 2Endoscopy Center, Osaka Medical and Pharmaceutical University, Osaka 565-0871, Japan

**Keywords:** PuraStat, bleeding, hemostasis, endoscopic sphincterotomy, ERCP

## Abstract

**Introduction:** Recently, a novel self-assembling peptide hemostatic gel has become available in Japan. However, the safety and efficacy of this novel self-assembling peptide hemostatic gel remain unclear for bleeding after EST. The aim of this study was to evaluate the safety and efficacy of a novel self-assembling peptide hemostatic gel for bleeding after EST, and to perform a comparison to a conventional endoscopic hemostasis technique. **Method:** This retrospective study was carried out between January 2019 and October 2022. Patients who developed bleeding associated with EST were enrolled. The patients were divided into two groups based on the hemostasis technique used: a conventional hemostasis technique (Group A) or a novel self-assembling peptide hemostatic gel hemostasis technique (Group B). **Result:** A total of 62 patients (Group A, *n* = 36; Group B, *n* = 26) were included. Endoscopic hemostasis was initially obtained in 72.2% (26/32) of patients in Group A and in 88.4% (23/26) of patients in Group B, which was not significantly different (*p* = 0.1320). However, the procedure time was significantly shorter in Group B (mean, 9.38 min) compared with Group A (mean, 15.4 min) (*p* = 0.0103). There were no significant differences in the severity of bleeding between the two groups (*p* = 0.4530). Post-EST bleeding was observed in six patients (Group A, *n* = 4; Group B, *n* = 2). Adverse events were more frequently observed in Group A (*n* = 12) than in Group B (*n* = 1) (*p* = 0.0457). **Conclusions:** PuraStat application for EST bleeding might be safe and effective, and is comparable to the conventional endoscopic hemostasis technique, although further prospective randomized trials are needed.

## 1. Introduction

Endoscopic sphincterotomy (EST) is an essential procedure before bile duct stone removal or metallic stent deployment under endoscopic retrograde cholangiopancreatography (ERCP) guidance. As an adverse event, bleeding is commonly observed; the frequency is 0.3–2% [1]. To obtain hemostasis, various techniques such as balloon tamponade, coagulation, hemoclip application, or epinephrine injection are traditionally applied [2,3,4]. As a novel and effective hemostasis technique, covered self-expandable metal stent (CSEMS) deployment has been reported [5]. Although endoscopic hemostasis may be obtained, there are concerns, such as the risk of post-ERCP pancreatitis, ulcer formation, and the high cost. Therefore, another technique that is safer and has better cost-effectiveness is needed.

Recently, a novel self-assembling peptide hemostatic gel has become available in Japan. Although this gel has been used for endoscopic mucosal resection in submucosal dissection [6,7,8], only a few case reports have been published in pancreato-biliary endoscopic procedures [9]. Therefore, the safety and efficacy of this novel self-assembling peptide hemostatic gel remain unclear. The aim of this study was to evaluate the safety and efficacy of a novel self-assembling peptide hemostatic gel for bleeding after EST, and perform a comparison to conventional endoscopic hemostasis technique.

## 2. Patients and Methods

This retrospective study was carried out at a single center between January 2019 and October 2022. As hemostasis strategies in our hospital, hemostasis techniques such as balloon tamponade, coagulation, clipping, or epinephrine injection were first performed from January 2019 to September 2021. Because a novel self-assembling peptide hemostatic gel was available in our hospital from October 2021, a novel self-assembling peptide hemostatic gel application was first performed between October 2021 and October 2022. In this study, bleeding during EST is classified as oozing, pulsatile, and projectile bleeding. The patients were divided into two groups based on the hemostasis technique used: a conventional hemostasis technique (Group A) or a novel self-assembling peptide hemostatic gel hemostasis technique (Group B). To compare the clinical efficacy between the two groups, because the novel self-assembling peptide hemostatic gel was not indicated for projectile bleeding, cases of projectile bleeding were excluded in this study.

All study protocols were approved by the institutional review board of our hospital. The study protocol conformed to the ethical guidelines of the 1975 Declaration of Helsinki as reflected in the a priori approval given by the human research committee at Osaka Medical College (IRB No.2021-019). The requirement for informed consent was waived due to the retrospective nature of this study.

### 2.1. Technical Tips for EST and Hemostasis Using Novel Self-Assembling Peptide Hemostatic Gel

Figure 1 shows the characteristics of the novel self-assembling peptide hemostatic gel (PuraStat^®^, 3D Matrix Europe SAS, Caluire-et-Cuire, France). PuraStat consists of a fully synthetic viscous peptide solution that forms a transparent hydrogel at neutral pH. After this gel is applied to a bleeding point, it rapidly forms a hydrogel barrier to obtain hemostasis. The contact between PuraStat and bodily fluids allows for the formation of a three-dimensional scaffold structure, rapidly converting the point of bleeding and providing a physical barrier and surface under which coagulation occurs. As result, hemostasis can be achieved.

A duodenoscope (TJF260V; Olympus, Tokyo, Japan) was inserted into the second part of the duodenum, and biliary cannulation was then attempted using a standard ERCP catheter (MTW Endoskopie, Düsseldorf, Germany). After a cholangiogram was obtained using contrast medium injection, a 0.025-inch guidewire (VisiGlide; Olympus, Madison, WI, USA) was deployed. Then, EST was performed using standard wire-guided sphincterotomes (Clevercut; Olympus) with a generator under an automatic cutout system (Endo-cut mode, ICC200; Erbe, Tübingen, Germany).

Technical tips for hemostasis using PuraStat are described as follows. First, the dedicated delivery catheter is filled by PuraStat at the top of the catheter (Figure 2a). Then, the bleeding point is carefully identified using saline injection. To prevent PuraStat dislocation into the third part of the duodenum, the duodenoscope is adjusted until the ampulla of Vater is at 3 to 6 o’clock on the endoscopic image (Figure 2b,c). Next, the dedicated delivery system is adhered at the bleeding point, and PuraStat is subsequently applied (Figure 2d,e). During this procedure, the duodenoscope is maintained in a stable position, and one must avoid injecting water or aspirating to prevent dispersion (Appendix A). However, this technique might be bothersome during bleeding because the endoscopic view might be poor due to bleeding. Therefore, sufficient water injection should be performed to detect the bleeding point and to obtain a clear endoscopic view before PuraStat application.

### 2.2. Definitions and Statistical Analysis

The post-EST bleeding was divided into three types: oozing, pulsatile, and projectile bleeding. Projectile bleeding was defined as bleeding with the presence of a blood projection, and pulsatile bleeding was defined as bleeding in the absence of a blood projection.

The severity of post-ES bleeding was defined according to the lexicon of the American Society for Gastrointestinal Endoscopy [10]. Moderate bleeding was defined as that requiring transfusion, intensive care unit admission, angiographic intervention, or prolonged hospitalization for 4 to 10 days. Severe bleeding was defined as that requiring surgical intervention, prolonged hospitalization for >10 nights, or an intensive care unit stay of >1 day. If post-ES bleeding did not correspond to any of these, it was defined as mild bleeding.

The management of antiplatelet and anticoagulant agents were as follows. Antiplatelet agents such as aspirin or thienopyridines were discontinued or replaced with heparin for 5–7 days before EST. Regarding anticoagulant agents, warfarin was discontinued for 5 days, and direct oral anticoagulants (DOACs) were discontinued for 2 days. Additionally, in cases with dual antiplatelet treatment, aspirin was continued and thienopyridine was discontinued for 5–7 days. After EST, antiplatelet and anticoagulant agents were restarted on the following day if bleeding was not complicated.

The main outcome of this study was the technical success rate for hemostasis. Adverse events associated with the application of PuraStat were also evaluated as secondary outcomes. Descriptive statistical data are presented as median (range) or mean (± standard deviation) and frequencies for continuous and categorical variables, respectively. The Mann–Whitney U test and Fisher’s exact test were used to compare quantitative variables. Differences with a *p* value < 0.05 were considered significant. Continuous variables are expressed as means. All data were analyzed mainly using SPSS version 13.0 statistical software (SPSS, Chicago, IL, USA).

## 3. Results

During the study period, a total of 4080 ERCP procedures was performed in our hospital. Among them, 694 patients underwent EST, and oozing and pulsatile bleeding were observed in 62 patients. Therefore, these 62 patients were included in this study. Table 1 shows the patients’ characteristics: 36 underwent a conventional hemostasis technique (Group A, median age, 77 years; 20 men) and 26 underwent the novel self-assembling peptide hemostatic gel hemostasis technique (Group B, median age, 77 years; 17 men). The number of antithrombotic agents (aspirin, thienopyridine, warfarin, or direct oral anticoagulants) was not significantly different between the two groups (*p* = 0.4202), and the management of antithrombotic agents, such as continuation/discontinuation or heparin replacement, was also not different (*p* = 0.4270). The main indication for EST was stone removal or biliary drainage, and there was no significant difference between the two groups (*p* = 0.3995). In addition, the mean platelet count (*p* = 0.2478) and PT-INR (*p* = 0.5733) were not significantly different between the two groups.

Table 2 shows the details of the procedures. Regarding the type of bleeding, oozing was observed in 24 patients in Group A and in 21 patients in Group B. Pulsatile bleeding was observed in 12 patients in Group A and in 5 patients in Group B (*p* = 0.2193). There were no significant differences in the severity of bleeding between the two groups (*p* = 0.4468). The procedure time was significantly shorter in Group B (mean, 9.38 min) compared with Group A (mean, 15.4 min) (*p* = 0.0103). Adverse events were more frequently observed in Group A (*n* = 12) than in Group B (*n* = 1). Mean hospital stay was significantly shorter in Group A (13.8 days) compared with Group B (7.8 days) (*p* = 0.0327).

Table 3 shows the results of endoscopic hemostasis. Endoscopic hemostasis was initially obtained in 72.2% (26/32) of patients in Group A and in 88.4% (23/26) of patients in Group B, which was not significantly different [odds ratio (OR) 2.95, 95% confidence interval (C.I) 0.72–12.04, *p* = 0.1320]. In the patients in Groups A (*n* = 10) and B (*n* = 3) who did not achieve initial hemostasis, coagulation or SEMS deployment was attempted, and hemostasis was finally obtained in all patients. After endoscopic hemostasis, post-EST bleeding was observed in six patients (Group A, *n* = 4; Group B, *n* = 2) (OR 0.52, 95% C.I 0.092–2.90, *p* = 0.453). In Group A, endoscopic hemostasis was initially attempted by SEMS deployment in all patients, but post-EST bleeding was observed. These patients underwent coagulation, and/or hemostasis was successfully achieved in all patients. In Group B, coagulation with SEMS deployment was attempted as a secondary hemostasis technique for two failed patients, and was successful.

## 4. Discussion

Bleeding is a well-known adverse event during and after EST. To obtain hemostasis, several techniques have been reported. Hypertonic saline–epinephrine (HSE) injection or coagulation is one of the useful techniques to obtain hemostasis for bleeding associated with EST. Schmitz et al. conducted a comparison study of epinephrine injection (*n* = 34), plastic stent deployment (*n* = 30), and both (*n* = 15) [11]. Clinical success rates for stopping post-EST bleeding using epinephrine injection, plastic stent deployment, and both were 97% (33/34), 100% (30/30), and 93% (14/15), respectively, with no significant difference. However, re-intervention was more frequent (*n* = 30 vs. *n* = 1; *p* < 0.001) and hospital stay was longer [median: 3 (2–10)] vs. 2 (1–3); *p* = 0.0357] in patients who underwent plastic stent deployment compared with epinephrine injection. Therefore, they concluded that epinephrine injection is a safe and effective technique. As another technique, Katsinelos et al. evaluated the efficacy and safety of endoscopically delivered monopolar coagulation through a polypectomy snare in patients with bleeding after EST not responding to injection treatment [12]. In this study, EST bleeding was observed in 59 of 672 consecutive patients (8.78%). Of them, in 11 patients with intraprocedural bleeding (7 oozing and 4 spurting) who did not respond to spraying irrigation and epinephrine injection, hemostasis was obtained by monopolar coagulation, with no adverse events. However, although severe adverse events were not noted in these studies, acute pancreatitis, ulcer formation, myocardial infraction or perforation can theoretically occur when these techniques are used.

Recently, SEMS deployment for bleeding associated with EST has been reported as a useful technique. Bilal et al. conducted a retrospective study including a relatively large patient cohort [13]. In their study, immediate EST bleeding was observed in 74 patients (76.3%), and delayed EST bleeding was observed in 23 patients (23.7%). SEMS deployment was attempted for 97 patients. Technical success was obtained in all patients. In addition, immediate hemostasis was achieved in all patients. Re-bleeding occurred in six patients (re-bleeding rate, 6%). Of these six patients, two underwent angiography, and one underwent surgical treatment. Of the other three patients, to underwent epinephrine injection, and one was successfully treated conservatively. Therefore, SEMS deployment may be useful as a hemostasis technique. However, the critical limitations of SEMS deployment for EST bleeding might be its high cost and the risk of acute pancreatitis due to pancreatic duct orifice obstruction caused by the SEMS.

Recently, PuraStat has been developed as a novel self-assembling peptide for hemostasis. Subramaniam et al. conducted a randomized, controlled trial to evaluate the reduction in heat therapy used in the PuraStat group compared with the control group during endoscopic submucosal dissection (ESD) [8]. In the 101 patients undergoing ESD, there was a significant reduction in the use of heat therapy for intraprocedural hemostasis in the PuraStat group compared with the control group (49.3% vs. 99.6%, *p* < 0.001), although there were no significant differences in procedure time, time for hemostasis, and the delayed bleeding rate. However, interestingly, complete wound healing at 4 weeks was higher in the PuraStat group (48.8%) than in the control group (25.0%; *p* = 0.02). Branchi et al. conducted a prospective, multicenter, observational study of the feasibility, safety, and efficacy of PuraStat for gastrointestinal bleeding [6]. In their study, 111 patients with upper gastrointestinal bleeding (70%, 78/111) and lower gastrointestinal bleeding (30%, 33/111) were included. Hemostasis using PuraStat was attempted as a first-line technique in 94% of these patients (74/79, 95% confidence interval 88–99%). In addition, the therapeutic success rates (absence of re-bleeding) after 3 and 7 days were 91% and 87%, respectively. Therefore, the application of PuraStat for gastrointestinal bleeding has clinical impact as a novel hemostasis technique. However, for pancreato-biliary endoscopic procedures, only a few case reports have been published [9,14]. The reason for this might be its technical difficulty. Due to anatomic reasons, if PuraStat is applied to the ampulla of Vater, it may be dislocated into the third part of the duodenum. Therefore, during this procedure, there are two important points. First, the duodenoscope should be adjusted until the ampulla of Vater is at 3 to 6 o’clock on the endoscopic image. If the applied PuraStat is normally dislocated into the third part, PuraStat should be applied from the upper side of the ampulla of Vater. Second, the duodenoscope should be maintained in a stable position, and one must avoid injecting water or aspirating to prevent dispersion. Indeed, because these techniques were used in the present study, PuraStat was applied successfully in all cases. PuraStat is effective as a hemostasis technique for EST bleeding based on the present results, and it was comparable to the conventional technique. The procedure time was significantly shorter, and adverse events were not observed in any patients, although there was no significant difference compared with the conventional technique. Interestingly, the hospital stay duration after initial hemostasis was significantly shorter in PuraStat group compared with conventional group. This reason might be based on the frequency of acute pancreatitis, and PuraStat application might be safe to prevent acute pancreatitis duce to endoscopic hemostasis. However, this explanation should be evaluated by further study.

This study had several limitations, such as that it was a retrospective, non-randomized, and single-center study. Moreover, the small sample size is a critical limitation; therefore, the results should be evaluated by a prospective, randomized trial with large sample size.

In conclusion, PuraStat application for EST bleeding might be safe and effective, and comparable to the conventional endoscopic hemostasis technique; however, our result should be evaluated by randomized controlled trials.

## Figures and Tables

**Figure 1 jcm-12-00079-f001:**
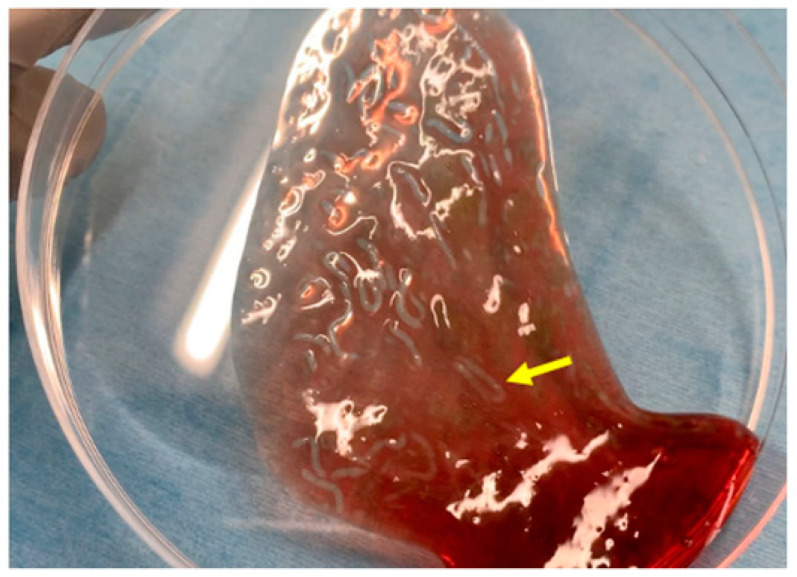
Images of PuraStat. After PuraStat injection into the blood, gel formation is observed (yellow arrow).

**Figure 2 jcm-12-00079-f002:**
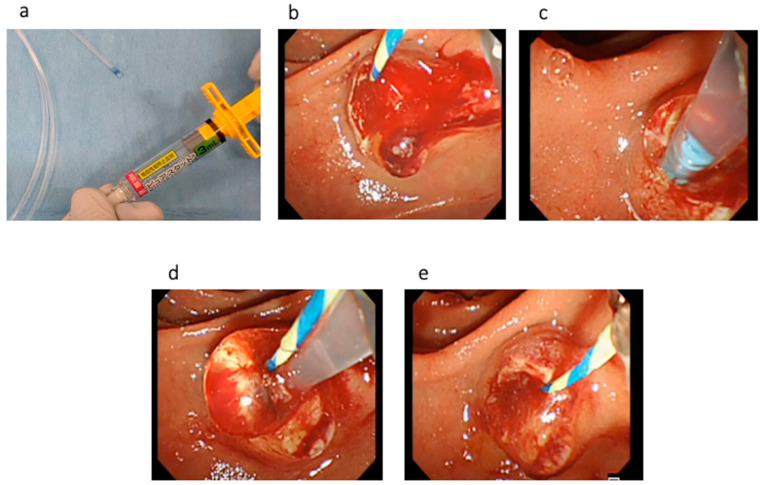
(**a**) The dedicated delivery catheter is filled with PuraStat at the top of the catheter. (**b**) EST bleeding is observed. (**c**) To prevent PuraStat dislocation into the third part of the duodenum, the duodenoscope is adjusted until the ampulla of Vater is at 3 to 6 o’clock on the endoscopic image. (**d**). PuraStat is subsequently applied. (**e**) Hemostasis is successfully obtained.

**Table 1 jcm-12-00079-t001:** Patient characteristics.

Characteristics	Group A (Conventional Group)	Group B (PuraStat)	*p*-Value
Total patients (*n*)	36	26	-
Median age (y, range)	77 (53–87)	77 (18–85)	0.6140
Sex (male/female)	20/16	17/9	0.4363
Antithrombotic agents, *n* Aspirin Thienopyridine Warfarin DOAC	3 1 2 1	5 1 0 2	0.4202
Management of Antithrombotic agents Continuation Discontinuation Heparin replacement	3 3 1	4 4 0	0.4270
Comorbidity, *n* (%) Liver cirrhosis Hemodialysis	1 2	0 2	0.6604
Indication for EST * Stone removal Biliary drainage Cholangioscopy Forceps biopsy Others	16 11 6 3 0	12 9 1 4 0	0.3995
Mean platelet count (10^4^/μL) (range)	21.9 (5–65)	23.5 (8.8–41.5)	0.2478
PT-INR ** (range)	1.20 (0.96–1.63)	1.19 (0.9–2.71)	0.5733

* EST: endoscopic sphincterotomy; ** PT-INR: Prothrombin Time—International Normalized Ratio.

**Table 2 jcm-12-00079-t002:** Details of the procedure.

Characteristics	Group A (Conventional Group)	Group B (PuraStat)	*p*-Value
Type of bleeding Oozing Pulsatile	24 12	21 5	0.2193
Mean procedure time (range)	15.4 (5–35)	9.38 (5–20)	0.0103
Type of initial hemostasis technique PuraStat Balloon tamponade Coagulation SEMS * deployment HSE ** injection	0 24 9 1 2	26 0 0 0 0	<0.0001
Type of combined hemostasis technique PuraStat Balloon tamponade Coagulation SEMS deployment	0 0 5 5	0 0 1 2	0.9472
Severity of bleeding Severe Moderate Mild	0 5 31	0 2 24	0.4468
Adverse events Acute pancreatitis Cholangitis Perforation Ulcer formation around Vater Cholecystitis	9 0 2 1 0	1 0 0 0 0	0.0457
Transfusion after initial hemostasis	3/31	1/26	0.4779

* SEMS; self-expandable metal stent, ** HSE; Hypertonic saline–epinephrine.

**Table 3 jcm-12-00079-t003:** Results of endoscopic hemostasis.

	Group A/Group B	Odds Ratio (95% Confidence Interval)	*p*-Value
Initial technical success rate, %	72.2 (26/36)/88.4 (23/26)	2.95 (0.72–12.04)	0.132
Rate of re-bleeding, % (*n*)	11.1 (4/36)/7.7 (2/26)	0.52 (0.092–2.90)	0.453

## Data Availability

Not applicable.

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
