# Peer review of "Comparison of Endoscopic Hemostasis for Endoscopic Sphincterotomy Bleeding between a Novel Self-Assembling Peptide and Conventional Technique"

_jcm, 2022, doi:10.3390/jcm12010079_

Round 1

Reviewer 1 Report

The authors of the manuscript “Comparison of endoscopic hemostasis for endoscopic sphincterotomy bleeding between a novel self-assembling peptide 3 and conventional technique” have evaluated the role of PuraStat, a hemostatic gel, for the management of post EST bleeding in a retrospective study. This is a nice concept as post-EST bleed can be troublesome at times, and use of coagulation devices or SEMS can trigger the risk of another iatrogenic issue of post-ERCP pancreatitis.

I have the following comments to make:

1)      The sample size is small. Although, it does address a key issue of post EST bleed and might throw light on the future prospects of use of PuraStat as a hemostatic agent for the same.

2)      The protocol followed after the availability of PuraStat is not clear in the methodology. Did all bleeds undergo PuraStat application at first, or did they use conventional techniques first? The authors have used the term “step-up” hemostasis strategy. What is meant by that?

3)       The technique of application of PuraStat appeared slightly industrious. Do the authors think that the need for a specific angulation of the vision and scope placement is not bothersome when there is ongoing bleed? Additionally, bleed can often obscure vision, and not being able to use water during gel application would make this difficult. Moreover, generalizability of using this gel with such stringent application technique seems dubious.

4)      In the outcomes table (Table 2), the comparison p values for a lot of outcomes have not been provided. Please provide p values and also CI for the various proportions. 

Author Response

Response to Reviewer 1 Comments

Point 1: The sample size is small. Although, it does address a key issue of post EST bleed and might throw light on the future prospects of use of PuraStat as a hemostatic agent for the same.

Response 1: Thank you for your comment and suggestion. I comletely agree with your opinion. Although sample size of our study is small, bleeding after EST itself may not be so many. Therefore, our result may be landmark for further trial. However, this sample size is critical limitation, and we modified limitation sentences in discussion section. Also, conclusion was changed.

Point 2: The protocol followed after the availability of PuraStat is not clear in the methodology. Did all bleeds undergo PuraStat application at first, or did they use conventional techniques first? The authors have used the term “step-up” hemostasis strategy. What is meant by that?

Response 2: We are sorry for this insufficient description. PuraStat was availabe in our hospital from October 2021. Therefore, between January 2019 and September 2021, conventional hemostasis technique was firstly applied, and between October 2021 and October 2022, PuraStat was firstly performed. In the latter technique was failed, conventional technique application was conisered. We added this in the ‘Definitions and Statistical analysis’ section. This means step-up, but this will be lead to misread, therefore, we deleted.

Once again, we are sorry for this insufficient description.

Point 3: The technique of application of PuraStat appeared slightly industrious. Do the authors think that the need for a specific angulation of the vision and scope placement is not bothersome when there is ongoing bleed? Additionally, bleed can often obscure vision, and not being able to use water during gel application would make this difficult. Moreover, generalizability of using this gel with such stringent application technique seems dubious.

Response 3: We completely agree with your opinion. As you pointed out, our technique may be industrioous. In fact, during bleeding, a specific angulation of the vision and scope placement is bothersome. However, in our study, PuraStat application is indicated for oozing or pulsatile bleeding. Therefore, endoscopic view may not be so poor. In addition, before PuraStat application, as you pointed out, we injected water to obtain clear endoscopic view. However, after PuraStat application, if water injection was perfomred, dislocation of PuraStat can occur. We corrected technical tips in the TEXT.

I hope you understand this explanation, and thank you again for your valuable opinions.

Point 4: In the outcomes table (Table 2), the comparison p values for a lot of outcomes have not been provided. Please provide p values and also CI for the various proportions.

Response 4: Thank you for yoursuggestion, and we sorry for this mistake. We added P-value in the Table and odds ratio and 95% CI were also added in Table 3.

  Finally, thank you for your valuable comments and suggestions.

Reviewer 2 Report

1.       The manuscript could be dramatically improved if the authors defined the total number of patients undergoing endoscopic sphincterotomy during this time period, 1/19–10/22, and whether all of the patients in this series had immediate vs delayed (post-procedure) bleeding. The latter is a more significant problem than a mild procedural ooze, almost all of which stop spontaneously.

2.       The authors need to differentiate pulsatile from projectile bleeding in the manuscript as PuraStat™ is not indicated in the latter.

3.       Table 1

a.       Define timing of continuation or discontinuation for management of anti-thrombotic agents.

b.       Define HSE. Tables should stand on their own.

c.       There is a statistically significant longer stay after initial hemostasis in the conventionally treated group. Please discuss reasons. ? Difference in co-morbidities. ? Post-ERCP pancreatitis.

4.       Define “myocardial infection” with conventional treatment for post-ES bleeding.

5.       How did the authors select patients to be in group A or B? Please clarify whether there were any patients in whom conventional Rx for post-ES bleeding was attempted once PuraStat™ became available. Were patients Rx with conventional Rx which failed treated with PuraStat™?

Author Response

Response to Reviewer 2 Comments

Point 1: The manuscript could be dramatically improved if the authors defined the total number of patients undergoing endoscopic sphincterotomy during this time period, 1/19–10/22, and whether all of the patients in this series had immediate vs delayed (post-procedure) bleeding. The latter is a more significant problem than a mild procedural ooze, almost all of which stop spontaneously.

Response 1: We completely agree with your suggestion. We added the total number of patients who underwent EST during this study period, and the frequency of post-EST bleeding. In addition delayed (post-procedure) bleeding was already described in the Table 2.

Thank you for your suggestion.

Point 2: The authors need to differentiate pulsatile from projectile bleeding in the manuscript as PuraStat™ is not indicated in the latter.

Response 2: Thank you for your suggestion. We previously described the kinds of bleeding during EST such as oozing, pulsatile and projectile. To compare two groups, projectile bleeding cases were excluded in this study. However, this may be unclear. Therefore, this explanation was moved on patients and method section, and modified. Once again, thank you for your valuable suggestion.

Point 3: Table 1

  1. Define timing of continuation or discontinuation for management of anti-thrombotic agents.
  2. Define HSE. Tables should stand on their own.
  3. There is a statistically significant longer stay after initial hemostasis in the conventionally treated group. Please discuss reasons. ? Difference in co-morbidities. ? Post-ERCP pancreatitis.

Response 3: a. We added timing of continuation or discontinuation for management of anti-thrombotic agents in the main TEXT.

  1. We added explanation of EST, SEMS, PT-INR, and HSE in the Table. 
  2. Thank you for your comment. This might be based on the frequency of acute pancreatitis. We discussed this in the TEXT.

Point 4: Define “myocardial infection” with conventional treatment for post-ES bleeding.

Response 4: We strongly sorry for this mistake. This word should be changed to myocardial infraction. Once again, we are sorry for this mistake.

Point 5: How did the authors select patients to be in group A or B? Please clarify whether there were any patients in whom conventional Rx for post-ES bleeding was attempted once PuraStat™ became available. Were patients Rx with conventional Rx which failed treated with PuraStat™?

Response 5: We are sorry for this insufficient description. PuraStat was availabe in our hospital from October 2021. Therefore, between January 2019 and September 2021, conventional hemostasis technique was firstly applied, and between October 2021 and October 2022, PuraStat was firstly performed. In the latter technique was failed, conventional technique application was conisered. We added this in the ‘Definitions and Statistical analysis’ section. This means step-up, but this will be lead to misread, therefore, we deleted.

Once again, we are sorry for this insufficient description.

  Finally, thank you for your valuable comments and suggestions.

Round 2

Reviewer 2 Report

1.       Please define the difference between pulsatile and projectile bleeding. They both seem to be arterial to the reviewer. Do you mean with projectile bleeding that the patient has hematemesis? As projectile bleeding precludes treatment with PuraStat, this is an important definition in this study.

2.       Do the authors mean that hospitalization time was shorter (rather than longer) in the Purastat- treated group, possibly related to the higher incidence of pancreatitis in group A? Please correct and add length of hospitalization to the Results section in the text or Table 3 if you are going to comment on it in the Discussion.

Author Response

Response to Reviewer 2 Comments

Point 1: Please define the difference between pulsatile and projectile bleeding. They both seem to be arterial to the reviewer. Do you mean with projectile bleeding that the patient has hematemesis? As projectile bleeding precludes treatment with PuraStat, this is an important definition in this study.

Response 1: We are sorry for this insufficient description. Projectile bleeding was defined as bleeding with the presence of a blood projection, and pulsatile bleeding was also defined as bleeding the absence of a blood projection. Therefore, pulsatile bleeding may be able to stopped by compressing the bleeding point with a catheter and PuraStat application. We added these sentences in method section. Once again, thnak you for your valuable comments and suggestions.

Point 2: Do the authors mean that hospitalization time was shorter (rather than longer) in the Purastat- treated group, possibly related to the higher incidence of pancreatitis in group A? Please correct and add length of hospitalization to the Results section in the text or Table 3 if you are going to comment on it in the Discussion.

Response 2: We are sorry for this mistake. Hospital stay was shorter in PuraStat group. We added this in the result section, and corrected discussion section. We would like to appriciate your suggestions and opinions.